# Syllogistic Reasoning and Knowledge Discovery

## Abstract

Though syllogistic reasoning is the most widespread and most well-known reasoning, its machine implementation is still an open problem. The widespread modus ponens formalization does not fully capture every reasoning steps that required to reach a syllogistic conclusion. This paper demonstrates that more information and knowledge discovery are necessary to complete a syllogistic reasoning. This paper also demonstrates how a knowledge discovery could be integrated into a syllogistic reasoning.

## Introduction

Syllogistic reasoning is a diverse reasoning family, which can be found in most of logical reasoning. The faculty of knowledge discovery plays a crucial role in syllogistic reasoning, but it is not demonstrated in most of formalizations. The reason is the knowledge discovery is something indemonstratable. This paper analyzes four typical syllogistic reasoning by causal analysis to show how the knowledge discovery participated in complete reasoning.

## A simple syllogistic reasoning

We say "A is B" and "B is C", then we conclude "A is C". Then the essential nature of syllogism is the formal cause (~>) of this demonstration, the formal cause introduces term (1) and (2), like:

$$\text{syllogism} \quad \sim> \quad \begin{array}{l} \text{(1) A is B} \\ \text{(2) B is C} \end{array}$$

The conclusion is:

$$\text{(3) therefore, A is C}$$

Obviously, the middle term is B. The substitution demonstration and transform is from (1)B to (2)B:

$$\text{(4)} \quad \text{"(1)B} \rightarrow \text{(2)B"}$$

It is a self-evident demonstration, because the middle-term transform "(1)B → (2)B" is self-contained. All knowledge required are already exist in (1) and (2). Whilst, the (1) and (2) are introduced by a common formal cause.

However, this simple syllogism is not a scientific syllogism, also it does not involves knowledge discovery. A scientific syllogism must contains something indemonstrable, this indemonstrable is grasped by knowledge discovery.

## A widespread syllogism

By contrary, the widespread syllogism involves noetic knowledge and needs knowledge discovery, as it is not self-sufficient.

At first, we may designate a formal cause (syllogism ~>) for this syllogism, which initially only contains two terms (premises): (5) and (6).

$$\text{syllogism} \quad \sim> \quad \begin{array}{l} \text{(5) all men are mortal} \\ \text{(6) Socrates is a man} \end{array}$$

$$\text{(7) Therefore, Socrates is mortal}$$

To conclude term (7), there must exist an (extra) intellectual transform, which is a progression from universal to particular:

"all men" (5) ------➤ "a man" (8)
"all men are mortal" (5) ------➤ "a man is mortal" (9)

Either (8) and (9) transformation is practically able to complete the whole syllogism, the (9) is more rational.

The (5) term exhibits an extensional semantics in this "syllogism". The (5) also exhibits as intensional semantics in a unity caused by nature. So, there must exist a unity of "all men" that consists of "man".

It is formally caused by the essential nature (nature ~>) of this unity, as below:

$$\text{Nature} \sim> \quad \text{all men} \rightarrow (\text{man, man, man, ... })$$

Then, the "(5) to (9)" transform must be done within the "all men" unity. The following diagram elaborates:

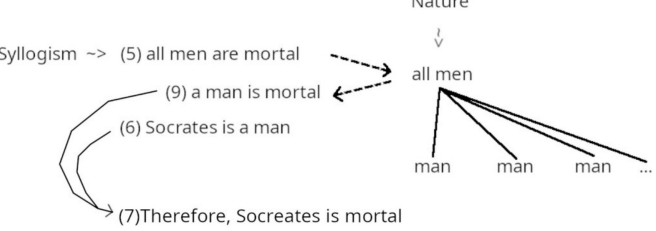

Figure 1: discovery in a widespread syllogism

There are two independent formal causes, syllogism and nature.

From term(5) to term(9), this is an efficient cause, illustrated as dash line. Term(9) is an efficient premise. Then, syllogistic conclusion (7) is a direct result produced by term(6) with term(9), rather than term(6) and term(5). The middle term between (6) and (9) is "a man". By contrary, there is no middle term between term(5) and term(6).

For the faculty of knowledge discovery, the transform "from (5) to (9)" is understood as "intellect". It is a progression from a nature unity to a syllogism. We say this crossing scope transform is an efficient cause. This efficient cause transforms a "all men" genus-species to term (9). This transform is grasped by knowledge discovery.

## The *Barbara* syllogism

The *Barbara* is the original syllogistic reasoning from Aristotle. It is shown as:

Barbara ~>     (14) all B are A.
                    (15) all C are B.

                    (16) therefore, all C are A.

The "Barbara ~>" formal cause only introduces two terms: (14) and (15).

As the middle term is through B, then it should have a unity "all B".

This progression is from universal to particular:
(18) "all B" ------> "B"

This means it transforms (14) "all B are A" to a new term:

(17) B be A

The (17) is a first principle grasped by knowledge discovery. The final conclusion (16) is a direct production of (15) and (17), in which the middle term is "B".

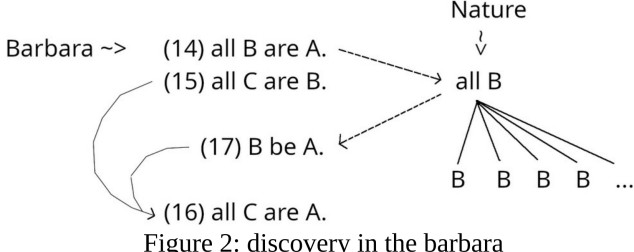

Figure 2: discovery in the barbara

In this *Barbara* reasoning. The term (17) is the real principle introduced by knowledge discovery.

## A complex case

The case is similar to the previous case. The difference is that the discovery is to do a job more like a jury duty. It also contains something indemonstrable.

(10) Everything with this essential nature has attribute X.
(11) Every instance of the genus has this essential nature.
(12) Therefore, every instance of the genus has attribute X.

To complete this syllogism, an additional transform (13) is needed the following transform is true:

(13) 'Every instance of the genus has the essential nature' 'Everything with this essential nature'

Semantically, the left-side of (13) is same as the right-side of (13), but its validity must checked by knowledge discovery. This transform also need to be done out of the syllogistic knowledge.

(10) Everything with this essential nature has attribute X
(11) Every instance of the genus has the essential nature
                                   genus
(13) Everything with this essential nature

Figure 3: discovery as jury duty

This syllogism is valid, if only if it satisfies "every instance of the genus has the essential nature". In other words, if only if "every instance of the genus has the essential nature" is

true. Whether it is true, it depends on the "genus". Because only the "genus" can decide (know) whether "every instance of the genus has" or "not has". If it is true then the (13) term could be returned, if not true then the (13) term does not exist. Also, the (13) is a first principle from knowledge discovery.

Similar to all above cases, the final conclusion (12) is a direct result of (10) and (13), but not (10) and (11). There is no direct middle term between (10) and (11).

## Conclusion

This paper provides a concise analysis regarding knowledge discovery in syllogistic reasoning. Though knowledge discovery is indemonstrable process for syllogistic reasoning, it is tractable through efficient cause. Also, it is able to be integrated into syllogistic reasoning as efficient premise. Knowledge discovery may be in diverse forms and implemented by various methods. When it is connected with syllogistic reasoning, this causal relationship is invariant.