# OpenReview forum: "Syllogistic Reasoning and Knowledge Discovery"
_AAAI.org/2025/Workshop/NeurMAD — AAAI 2025 Workshop NeurMAD Submission_

### Official Review · Reviewer_rt6x · 2024-12-25
**a short and clear written paper, a bit out-of dated**

**Rating:** 3
**Confidence:** 5

**Review:**

This paper sketches the idea that more information is necessary to complete syllogistic reasoning, and these necessary information shall be acquired through a process of knowledge discovery. Authors did not list any references.

Authors follow the early method of verbal analysis, and strictly distinguish "a man" from "all men". That is, from "a feature of all men" we can deduce "a feature of a man".  To deduce "Socrates has a feature", we shall first deduce "Socrates is a man". "All men" denotes a set, which corresponds to a predicate. This predicate applies for an instance. If its value is true, this instance is a member of the set, otherwise, this instance is not a member of the set. What authors propose is the need and the discovery of such a predicate.

In the analysis of the Barbara syllogism, authors suggest "all B" --> "B" and "all B are A"--> "B be A". In the term of set-theory, "all B are A" means that set B is a sub-set of set A, "B be A" means that set B equals to set A (let B be A). So, here, the analysis is incorrect (nor necessary).

---

### Decision · Program_Chairs · 2024-12-30

**Decision:**

Reject

**Comment:**

 This paper lacks a formal part (math) and necessary references to appear at this workshop.